# Comparing Apparent Diffusion Coefficient and FNCLCC Grading to Improve Pretreatment Grading of Soft Tissue Sarcoma—A Translational Feasibility Study on Fusion Imaging

**DOI:** 10.3390/cancers14174331

**Published:** 2022-09-05

**Authors:** Madelaine Hettler, Julia Kitz, Ali Seif Amir Hosseini, Manuel Guhlich, Babak Panahi, Jennifer Ernst, Lena-Christin Conradi, Michael Ghadimi, Philipp Ströbel, Jens Jakob

**Affiliations:** 1Department of General, Visceral and Pediatric Surgery, University Medical Center Goettingen, 37075 Goettingen, Germany; 2Institute of Pathology, University Medical Center Goettingen, 37075 Goettingen, Germany; 3Department of Diagnostic and Interventional Radiology, University Medical Center Goettingen, 37075 Goettingen, Germany; 4Department of Radiotherapy and Radiation Oncology, University Medical Center Goettingen, 37075 Goettingen, Germany; 5Department of Trauma Surgery, Orthopedics and Plastic Surgery, University Medical Center Goettingen, 37075 Goettingen, Germany; 6Department of Surgery, University Medical Center Mannheim, Medical Faculty Mannheim, Heidelberg University, 68167 Mannheim, Germany

**Keywords:** tumor heterogeneity, core needle biopsy, MRI, functional imaging, ADC

## Abstract

**Simple Summary:**

Histological subtype and grading are essential for the planning of soft tissue sarcoma. Pretherapeutic grading based on core needle biopsies is frequently not reliable due to intratumoral heterogeneity. This pilot study assessed the ability of functional radiological imaging to improve histopathological grading. Multiple biopsies were taken from the sarcoma specimens during tumor resection and radiopaque markers were placed. Subsequently, fusion of preoperative magnetic resonance imaging and postoperative computed tomography of the specimen allowed for comparison of histopathological grading and diffusion-weighted imaging. The apparent diffusion coefficient appears to correlate with FNCLCC criteria and may supplement pretreatment assessment and multimodal treatment allocation in soft tissue sarcoma.

**Abstract:**

Histological subtype and grading are cornerstones of treatment decisions in soft tissue sarcoma (STS). Due to intratumoral heterogeneity, pretreatment grading assessment is frequently unreliable and may be improved through functional imaging. In this pilot study, 12 patients with histologically confirmed STS were included. Preoperative functional magnetic resonance imaging was fused with a computed tomography scan of the resected specimen after collecting core needle biopsies and placing radiopaque markers at distinct tumor sites. The Fédération Nationale des Centres de Lutte Contre le Cancer (FNCLCC) grading criteria of the biopsies and apparent diffusion coefficients (ADCs) of the biopsy sites were correlated. Concordance in grading between the specimen and at least one biopsy was achieved in 9 of 11 cases (81.8%). In 7 of 12 cases, fusion imaging was feasible without relevant contour deviation. Functional analysis revealed a tendency for high-grade regions (Grade 2/3 (G2/G3)) (median (range) ± standard deviation: 1.13 (0.78–1.70) ± 0.23 × 10^−3^ mm^2^/s) to have lower ADC values than low-grade regions (G1; 1.43 (0.64–2.03) ± 0.46 × 10^−3^ mm^2^/s). In addition, FNCLCC scoring of multiple tumor biopsies proved intratumoral heterogeneity as expected. The ADC appears to correlate with the FNCLCC grading criteria. Further studies are needed to determine whether functional imaging may supplement histopathological grading.

## 1. Introduction

Treating of soft tissue sarcoma (STS) is a complex interdisciplinary task, as STS comprises more than 80 different subtypes and can occur at various sites in the entire body [1]. Treatment of low-grade sarcoma in localized stages is based on surgical resection, whereas neoadjuvant therapy concepts, such as chemotherapy, radiation, or both, must be considered for locally advanced high-grade tumors [2,3]. Therefore, the pretherapeutic assessment of tumor size, grade, and histological subtype is crucial for shared decision making and treatment allocation in STS [4].

The Fédération Nationale des Centres de Lutte Contre le Cancer (FNCLCC) established the most commonly used histopathological grading system based on three criteria: tumor differentiation, tumor necrosis, and mitotic count [5]. However, determining the FNCLCC score from pretreatment biopsies is not always reliable. Cohort studies demonstrate a concordance in grading of only 32% and 92.5% between pretreatment biopsies and resected specimens [6,7]. The significant differences in the accuracy of biopsies may be explained by the retrospective nature of the reports, missing standardization of tissue retrieval, choice of core needle size, and differences in biopsy guidance (none vs. ultrasound vs. computed tomography (CT)). Furthermore, tumor heterogeneity and failure to identify the most representative tumor site may also account for low accuracy [8]. The histopathological analysis from biopsies is based on a small tissue sample [6,9], although STS exhibits a distinct inter- and intratumoral heterogeneity and the mean tumor size ranges between 8 and 9.9 cm [10,11].

Some authors have suggested including functional imaging such as diffusion-weighted magnetic resonance imaging (DWI) to establish a radiological grading [12]. The apparent diffusion coefficient (ADC) assessment in DWI is based on Brownian molecular motion. Higher cellularity and restriction of molecular motion result in ADC decreases. Retrospective cohort studies have implied that differentiating malignant and benign soft tissue tumors may be feasible based on of the ADC [13]. Likewise, the correct differentiation into three stages of grading based on functional ADC imaging has previously been attempted [14]. Nevertheless, these and other cohort studies of functional imaging have not yet led to a reliable radiological grading system.

This pilot study aimed to learn more about the relation of radiological and histopathological grading by correlating functional imaging features with characteristics of histopathological assessment of distinct tumor sites. We used preoperative functional magnetic resonance imaging (MRI), intraoperative tumor biopsy and placement of radiopaque tissue markers, postoperative specimen CT, and fusion imaging to correlate ADC and FNCLCC criteria for distinct tumor sites.

## 2. Materials and Methods

### 2.1. Patient Population

The University Medical Center Goettingen ethics committee approved this prospective, open-label, single-center pilot study (application number: 2/9/19). All patients were treated at the Sarcoma Center Goettingen and selected based on the following inclusion criteria: age > 18 years, histologically confirmed STS, and scheduled tumor resection. All patients signed their informed consent.

### 2.2. Magnetic Resonance Imaging

Preoperative MRI was performed using 1.5 T and 3 T systems (MAGNETOM Symphony, MAGNETOM Vida, MAGNETOM Skyra) from Siemens Healthcare GmbH, Erlangen, Germany. The following sequences were included in the MRI protocol: axial T1-weighted sequence prior to contrast media application without fat suppression, sagittal T2-weighted half-Fourier acquisition single-shot turbo spin-echo imaging (HASTE) sequence with fat suppression, axial T2-weighted BLADE sequence with fat suppression, and axial and sagittal T1-weighted sequence with fat suppression after contrast media application. In addition, DWI, including the ADC sequence, was performed by an axial echo-planar imaging sequence with the b values 0, 50, and 800 s/mm^2^. Section thickness was set at 5 mm.

### 2.3. Intraoperative Core Needle Biopsies

Immediately after wide tumor resection, we collected core needle biopsies (CNBs) using the coaxial technique from two to four distinct non-necrotic tumor regions. Tumor areas with low ADC values were identified by an expert radiologist (A.S.) to identify suitable regions for intraoperative biopsies. These regions of interest (ROIs) served as target areas for intraoperative CNBs performed by the sarcoma surgeon (J.J.) (Figure 1). The CNBs were performed with a 12- to 14-gauge needle. Directly after the biopsy, a radiopaque marker was placed through the coaxial needle at each site. The samples were fixed in 4% formaldehyde. For all resected tumors in total and each CNB, grading according to FNCLCC criteria (tumor differentiation, tumor necrosis, and mitotic count) was determined by an expert pathologist (P.S.).

### 2.4. Computed Tomography Imaging

A native 128-slice fine-slice spiral CT (SOMATOM Definition Flash and SOMATOM Force from Siemens Healthcare GmbH, Erlangen, Germany) with multiplanar reconstruction was performed to visualize the biopsy markers from all resected specimens, each with a soft tissue and bone algorithm in the axial and sagittal layering (Figure 1). The section thickness ranged from 1 to 3 mm.

### 2.5. Fusion Imaging

Fusion imaging using the preoperative in situ MRI and postoperative CT of the specimens were performed with Eclipse software (v. 15.5 from Varian Medical Systems, Palo Alto, CA, USA). First, both the MRI and CT data sets were transferred into the software. The introduced biopsy markers were identified and color-coded in the CT of the specimens. An undirected fusion of the data sets was created automatically by the software. The orientation and localization of the CT image within the in situ MRI were visually adjusted and complemented by the software-assisted matching. The result is presented in Figure 2 and in Appendix A.

### 2.6. Image Analysis

The Centricity Universal Viewer (v. 6.0, GE Healthcare, Chicago, IL, USA) was used for the image analysis and calculation of the ADC values. In the in situ MRI, the ADC_mean_, ADC_min_, and ADC_max_ of the whole tumor were determined using an ellipsoidal measuring tool placed to capture as much of the tumor as possible. Fusion imaging allowed us to acquire the coordinates of all inserted biopsy markers on the in situ MRI scan. Consequently, we also determined the ADC values for all biopsy sites. For the measurement, a circular area with a diameter of 2 cm was defined around the coordinate point. This area corresponds to the approximate size of the biopsy punch radius during biopsy collection.

### 2.7. Statistical Analysis

Data are presented descriptively and the categories, proportions, median, and range were given. In addition, box plots were displayed if appropriate. The analyses were performed using IBM SPSS Statistics (v.28. IBM Corp., Armonk, NY, USA) and Microsoft Excel (v.16.53. Microsoft Corp., Redmond, WA, USA).

## 3. Results

### 3.1. Patient and Tumor Characteristics

Fourteen patients were prospectively included between January 2020 and March 2021. Of these, 12 patients were considered for the study evaluation. One case with a previous biopsy pointing toward sarcoma was excluded due to a benign histopathological diagnosis, and one case was excluded due to an inadequate MRI scan. Table 1 provides an overview of the patient and tumor characteristics.

The included well-differentiated liposarcoma was located deep in the fascia, measured 20 × 14 × 10 cm, originated in the adductor muscles, and grew into the ischiocrural muscles. In addition to these clinical criteria, imaging and histopathological analysis revealed necrotic areas within the tumor. Postoperatively, the resection with sparing of the sciatic nerve (R1) and a radiation therapy were performed. The solitary fibrous tumor (SFT) was also located deep to the fascia, had a size of 6.2 cm and originated in the fossa popliteal. It was classified to have a low risk of recurrence.

### 3.2. FNCLCC Score and Grading

The histopathological analysis of the entire specimens identified four Grade 1 (G1), six Grade 2 (G2), and one Grade 3 (G3) tumors (Table 2). Due to extensive regressive transformation after neoadjuvant treatment, in one case no grading could be determined (Gx). We used the determined FNCLCC score for each biopsy to derive the corresponding biopsy grade, as initially intended by the FNCLCC grading system for whole sarcoma specimens. Again, in one case, grading of the collected biopsies was not possible because of the extensive regressive transformation after neoadjuvant treatment of the entire tumor. For the cases with successful grading, there was a concordance in grading between the entire specimen and at least one biopsy (“best biopsy”) in 9 of 11 cases (81.8%). In the other 2 of 11 cases (18.2%), an “undergrading” (G1 instead of G2) occurred because the mitotic count in both cases and the tumor necrosis in one case could not be adequately determined based on the biopsies. Both cases were myofibroblastic sarcoma.

### 3.3. Fusion Imaging

In 7 of 12 cases (58.3%), fusion imaging was feasible without relevant contour deviation (Figure 2 and Appendix A), which allowed us to identify the precise position of the postoperatively placed markers in the preoperative MRI. For this purpose, the coordinates were recorded in the fusion software and later transferred to the MRI image for further analysis. In the other 5 of 12 (41.7%) cases, postoperative deformation prevented image fusion of sufficient quality. This difficulty occurred particularly in specimens with soft tumor consistency (4 of 5). Relevant contour deviations were observed in two cases of liposarcoma; one case of a myofibroblastic sarcoma; and one case of an undifferentiated sarcoma, not otherwise specified (NOS). In another case of a multifocal interjoint tumor (1 of 5), resection was performed through transfemoral amputation. The deviated position of the knee joint on postoperative specimen CT compared with the in situ MRI complicated a reliable fusion imaging. 

### 3.4. Correlation of the ADC and FNCLCC Score

A total of 23 pairs of ADC and FNCLCC scores at distinct tumor sites were available after fusion imaging. Figure 3 presents the correlation of ADC_mean_ with the corresponding grading. The mean ADC values (mean (range) ± standard deviation [×10^−3^ mm^2^/s]) were 1.43 (0.64–2.03) ± 0.46 for G1 and 1.13 (0.78–1.70) ± 0.23 for G2/G3. In addition, FNCLCC scoring of multiple tumor biopsies proved intratumoral heterogeneity as expected. Further, ADC appears to correlate with FNCLCC grading criteria. Due to the small study population, no significance analysis was performed. However, Figure 3 demonstrates the tendency of high-grade sarcoma regions (G2/G3) to have lower ADC values than low-grade (G1) sarcoma regions. Figure 2 and Appendix A present the fusion imaging of a retroperitoneal leiomyosarcoma. In addition, Figure 4 displays the direct correlation of ADC_mean_ values and FNCLCC grading of the different biopsies from the tumor. This example demonstrates the complexity of the histopathological detection of the mitotic count based on CNBs and illustrates the potential benefit of functional imaging in this context.

## 4. Discussion

Grading is one of the most important criteria for treatment allocation in patients with sarcoma. However, grading is a rather unreliable parameter in the pretreatment assessment of STS. We conducted this pilot study to learn more about the correlation between functional imaging and histopathological features of distinct tumor sites to improve the pretreatment assessment of STS.

The FNCLCC scoring system is currently the most used method for assessing the histopathological grading of STS. To calculate the FNCLCC score, tumor necrosis, tumor differentiation, and mitotic rate are determined. The method itself was developed and validated in the 1980s to 1990s based on cohort studies in France [15]. In these studies, pathologists carefully examined entire specimens of unpretreated sarcomas to develop a reliable and robust grading system to estimate the prognosis of STS. Today, however, pretreatment grading is usually determined based on biopsies rather than specimens, and perioperative treatment algorithms have become more sophisticated, e.g., grading is one of the most important triggers of neoadjuvant chemotherapy. This situation leads to two major clinical difficulties: First, the application of preoperative therapies may significantly alter FNCLCC parameters, such as tumor necrosis and mitotic count. Thus, postoperative re-evaluation and definitive determination of grading are frequently not possible at all. Second, grading based on biopsy material is consistently unreliable. Cohort studies comparing the grading of biopsies and specimens exhibit a variable overall concordance of 30% to 90% [6,7,9]. Some authors have argued that the unreliability of biopsies depends on the biopsy technique, favoring incisional biopsies over CNBs. Nevertheless, such an advantage of incisional biopsies has never been proven in head-to-head trials, and numerous tumors (e.g., retroperitoneal sarcoma) are unsuitable for incisional biopsies [16]. Intratumoral heterogeneity is the most crucial reason for the disappointing accuracy of pretreatment interventional biopsies [6,9].

In this study, we deliberately biopsied vital tumor sites at different sites but did not correctly estimate the grading in more roughly 10% of the cases. To our knowledge, better results were achieved only in a single study in which tumor biopsies were taken immediately during MRI [17]. With this method, Noebauer et al. achieved a reliable grading in 90.5% of cases. In clinical practice, however, this procedure is neither common nor practicable.

As cross-sectional imaging is an essential staging feature, assessing tumor characteristics using radiological methods is a logical approach to improving pretreatment grading. In particular, functional imaging may be suitable to describe tumor aggressiveness by estimating surrogate metabolism markers, blood flow, and tumor cellularity. Cohort studies have described MRI, CT, and positron emission tomography (PET)-CT as applicable modalities for this purpose [18,19,20].

The results of several cohort studies evaluating DWI suggest that ADC, as a parameter of cellularity, has the potential to characterize dignity and grading in STS [13,21,22]. Tumor necrosis as a surrogate marker of fast and ineffective tumor growth may be estimated using CT [18]. Mcaddy et al. applied the grading system according to FNCLCC but replacing histopathologic diagnosis of tumor necrosis with the CT imaging results. This method improved the sensitivity of necrosis detection, and subsequently, the reliability of grading. Metabolic activity measured using ^18^F-fluorodeoxyglucose (FDG)-positron emission tomography (PET)/CT also seems to be a potential functional parameter differentiating between high- and low-grade sarcomas. Several studies have recently demonstrated significant results for grading differentiation based on the standardized uptake value (SUV). Additionally, a correlation between the mitotic count or tumor necrosis and the SUV could be demonstrated here [20,23,24].

In the present work, DWI was chosen as a functional parameter for radiological tumor characterization. In addition, DWI may be suitable for estimating the dignity of unclassified soft tissue lesions and the grading of malignant STS. Chhabra et al. and Razek et al. correlated grading and ADC in retrospective cohort studies and found significantly lower ADC values for high-grade tumors than for low-grade tumors [14,25]. Similar results have also been demonstrated for other tumor entities, such as breast cancer [26]. Moreover, the DWI method provides advantages regarding clinical feasibility. It does not require additional radiation exposure, unlike CT or FDG-PET. Regarding cost-effectiveness, DWI is a convenient approach because MRI imaging is part of the standard diagnostics in STS and DWI is already established as a standard procedure in diagnosing for other indications such as ischemic stroke [27]. Therefore, its comprehensive feasibility should be ensured.

This pilot study has limitations. The chosen method of fusion imaging, intraoperative biopsy, and specific correlation of histology and functional imaging is complex and was realized only in a few patients. The results regarding the assumed correlation of grading and ADC must be validated in larger cohorts. In these large-scale validation studies, the appropriate consideration of the already-existing, subtype-specific scoring systems for risk assessment, such as the SARCULATOR or SFT prognosis tool, in addition to the general FNCLCC grading system, will be a specific challenge [28,29].

Furthermore, the technique of fusion imaging itself was limited by tumor deformation in this pilot study. Surgical manipulation leads to tumor deformation. This deformation must be compensated for with the appropriate software during image fusion [30]. We have chosen a commercially available software that was initially developed for image fusion of different imaging modalities to facilitate necessary calculations for radiotherapy. Due to the nature of the pilot study, in-house development of an appropriate program or adaptation of software approaches using machine learning or artificial intelligence was not reasonably possible. This development would have required a significantly higher number of patients to form training and test cohorts. For example, Breininger et al. used over 60 data sets to develop image fusion methods in the context of endovascular aortic prosthesis implantation [31]. Sarcomas are rare, vary in localization, and are sometimes mobile concerning neighboring structures. For more common cancers, which are frequently fixed and almost always occur at the same site, fusion imaging has already been developed for clinical practice, e.g., for MRI and ultrasound in the case of prostate cancer [32]. For sarcoma, such an approach is probably much more challenging.

As described, other authors have also pursued functional imaging approaches to improve the preoperative grading of sarcomas. Crombé et al. proposed a radiological grading system based on three independent radiological MRI criteria: peritumoral enhancement, heterogeneous signal intensity of 50% at T2-weighted imaging, and presence of necrotic tumor areas were associated with high-grade STS [12]. Peeken et al. and Navarro et al. also developed complex MRI-based artificial intelligence methods to differentiate low- and high-grade sarcomas [33,34]. However, none of these approaches achieved a more consistent quality than the current assessment of histological grading. The accuracy results in predicting high-grade tumors were 70.2%, 64.0%, 78.0%, and 83.0% [12,33,34]. Retrospective trials correlating radiographic and histological tumor features with oncological outcome are limited by patient selection, missing methodological standardization, and missing site-specific correlation of histology and imaging features. A large proportion of patients are referred to sarcoma centers with MRI of good quality but without functional imaging. These patients would not be eligible for further analyses since their imaging is usually sufficient for routine treatment planning. In the remaining cases, interventional biopsies are usually taken in a fan-shaped manner, and the exact site of the single biopsy is not documented. Other authors and guidelines recommend taking tumor heterogeneity into account and choosing the tumor area with the highest grade for biopsy [2,3,17,35]. Yet, no validation study ever demonstrated a reliable correlation of radiographic and histopathological tumor features. The rationale of this feasibility study was to establish a method of fusion imaging and the correlation of radiographic and histopathological parameters to prepare a prospective, large-scale validation study.

We considered two other methodological options for this prospective trial to enable a site-specific correlation of radiological and pathological features. The first option was to place a radiopaque marker at the biopsy site in situ, analogous to those used in biopsies of breast cancer [36]. However, these markers are currently not licensed for soft tissue lesions in Germany. Furthermore, additional postinterventional imaging for image fusion would also have been required. The second option was to perform CT-guided biopsies in all patients. Subsequently, the biopsy tracts (or the position of the coaxial needle) could have been correlated with preexisting functional imaging. Both alternatives require the conduction of feasibility studies to calculate the size of the study population, the setting of the trial, the number of study centers, etc., for a large-scale trial. Furthermore, for both alternatives, the development of a fusion imaging method would have been necessary to correlate biopsy sites and functional imaging. In addition, radiation exposure would have been increased—at least in comparison to our own routine approach—in which ultrasound-guided biopsies are preferred.

We believe that the most reasonable approach for a future grading system is a combination of histopathological and radiological assessment. In some tumors, such as angiosarcoma, the histological subtype in itself determines the biological aggressiveness and grading is not performed [1]. In the remaining cases, imaging can highly likely describe the tumor necrosis parameter better than a single biopsy from the tumor. As another parameter of the scoring system, the mitotic rate is a surrogate of tumor proliferation, which may be described by hypervascularization or cellularity, which are parameters of functional imaging (see Figure 4). Thus, the goal must be to develop a new scoring system incorporating a histological and molecular biological examination of the biopsy and functional imaging parameters (Figure 5).

## 5. Conclusions

In conclusion, we demonstrated the feasibility of fusion imaging of preoperative in situ MRI and postoperative CT of the specimen. Considering that the pretherapeutic grading is fundamental for treatment planning of STS, we demonstrated in selected cases that functional MRI could be a valuable complement to histopathological grading. In the future, a combined grading system of histopathological and radiological features could be developed to improve the reliability of pretherapeutic grading.

## Figures and Tables

**Figure 1 cancers-14-04331-f001:**
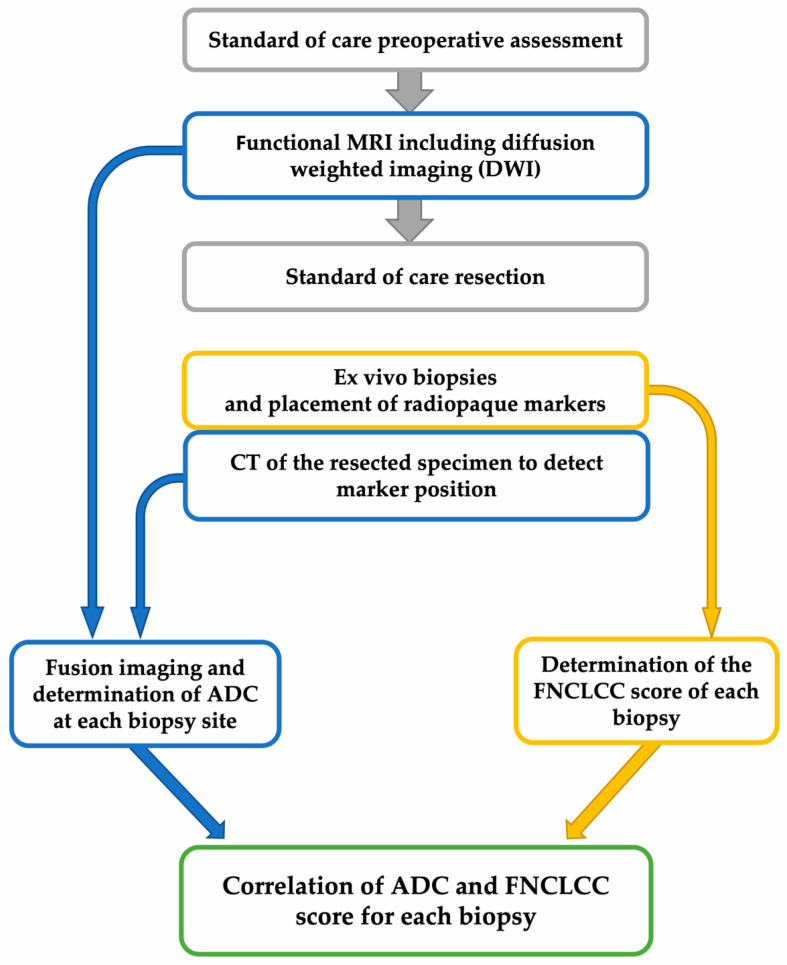
Study workflow. CT: computed tomography; ADC: apparent diffusion coefficient; FNCLCC: Fédération Nationale des Centres de Lutte Contre Le Cancer.

**Figure 2 cancers-14-04331-f002:**
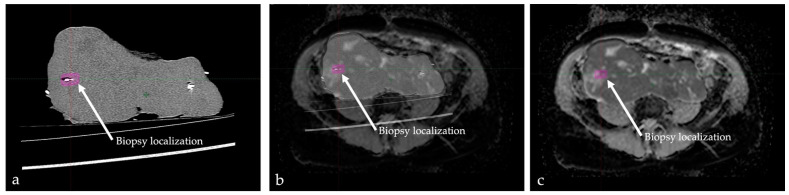
Fusion imaging of a leiomyosarcoma of the vena cava. (**a**) Postoperative CT of the specimen. One inserted biopsy marker is highlighted in pink (arrow). (**b**) Fusion sequence of postoperative CT and ADC sequence of the preoperative magnetic resonance imaging (MRI). (**c**) Complete fusion imaging with high tumor contour accuracy between CT and MRI. The biopsy marker visualizes the precise biopsy location within the preoperative in situ MRI. Please also consider the dynamic fusion in Appendix A.

**Figure 3 cancers-14-04331-f003:**
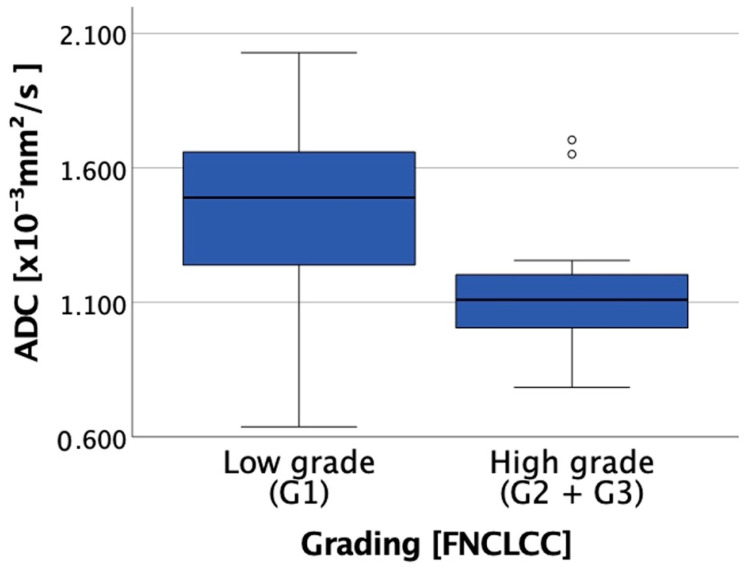
Boxplot showing the correlation between ADC and grading (low grade (Grade (G) 1) vs. high grade (G2 and G3)).

**Figure 4 cancers-14-04331-f004:**
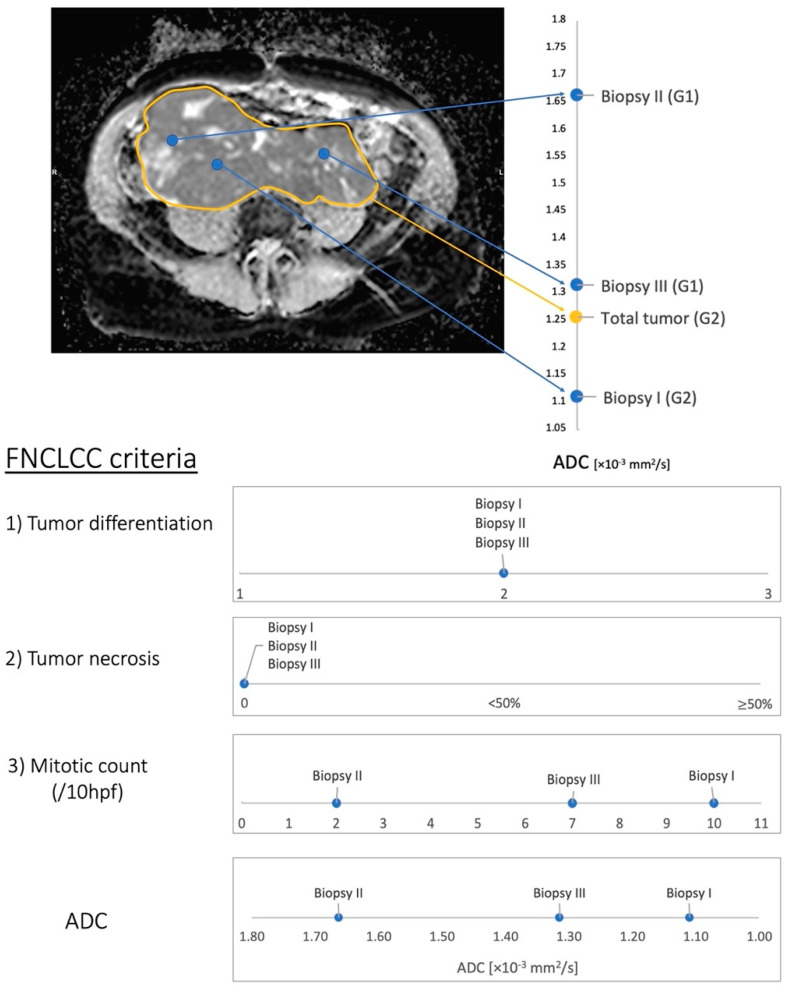
Distribution of the ADC of the entire tumor and the three collected biopsies. The arrows mark the distribution of ADC from the entire tumor and the biopsies. The ADC of biopsy III and the entire tumor are very close to each other. However, the grading reveals a discrepancy: total tumor: grade (G) 2, biopsy III: G1, because this is the lower mitotic count of biopsy III determined by histopathological examination. Histopathological assessment of the whole tumor revealed an FNCLCC grade of 2. The ADC_mean_ of the entire tumor was 1.26 ± 0.28 ×10^−3^ mm^2^/s, which was low and indicated a high grade. Biopsy I also revealed an FNCLCC score corresponding to G2 and low ADC values. Biopsy II revealed an FNCLCC score corresponding to G1 and high ADC values. At both intratumoral sites, the correlation of the ADC and FNCLCC score was consistent (high grade—low ADC; low grade—high ADC). The difference in grade and ADC reflected intratumoral heterogeneity. A discrepancy occurred between the histopathological and radiological findings at the third biopsy site. The ADC of Biopsy III was 1.32 ± 0.21 × 10^−3^ mm^2^/s, which is low and indicates a high grade. The FNCLCC score revealed a grade of 1. Comparing the single criteria of the FNCLCC score of each biopsy, all three biopsies were identical in terms of histological subtype and tumor necrosis but different in the number of mitoses, which was decisive for the grading determination. Thus, the critical difference was the mitotic count, which was 2 (G1–ADC high), 7 (G1–ADC low), and 10 (G2–ADC low; see Figure 3).

**Figure 5 cancers-14-04331-f005:**
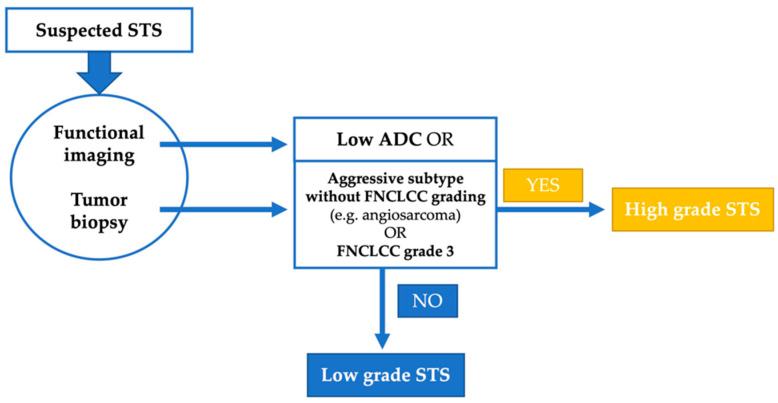
Suggestion of a combined grading system for soft tissue sarcoma (STS).

**Table 1 cancers-14-04331-t001:** Patient and tumor characteristics.

	Variable	*n* or Median (Range)
Patient	Number of patients	12
characteristics	Age (years)	72 (45–79)
	Sex: Female/Male	5/7
	Ethnicity: Caucasian/Other	12/0
Tumor characteristics	Primary tumor	10
	Recurrent tumor	2
	Tumor size (cm)	11.25 (6.2–22)
Histologic subtype	Well-differentiated liposarcoma	1
	Dedifferentiated liposarcoma	2
	Leiomyosarcoma	1
	Synovial sarcoma	1
	Myofibroblastic sarcoma	2
	MPNST	1
	Myxofibrosarcoma	1
	Solitary fibrous tumor	1
	Undifferentiated Sarcoma, NOS	2
Tumor localization	Upper extremity	3
	Lower extremity	5
	Intraabdominal	2
	Retroperitoneum	2
Grading total tumor	Grade 1 (G1)	4
(FNCLCC)	Grade 2 (G2)	6
	Grade 3 (G3)	1
	Grade x (Gx)	1

MPNST: Malignant peripheral nerve sheath tumor; NOS: not otherwise specified.

**Table 2 cancers-14-04331-t002:** Histological subtype and grading according to FNCLCC criteria of all total tumors and biopsies. The biopsies graded in concordance with the total tumor are highlighted in bold.

Histologic Subtype	Total Tumor					Biopsies				
	TD	TN	MC	FNCLCC Score	Grading	TD	TN	MC	FNCLCC Score	Grading
Dedifferentiated liposarcoma	1	0	<10	2	G1	B1: x	B1: 100%	B1: x	B1: x	B1: x
						B2: 1	B2: 0	B2: 0	B2: 2	**B2: G1**
						B3: 1	B3: 0	B3: 0	B3: 2	**B3: G1**
						B4: x	B4: 100%	B4: x	B4: x	B4: x
Synovial sarcoma	2	0	<10	3	G1	B1: 2	B1: 0	B1: 0	B1: 3	**B1: G1**
						B2: 2	B2: 0	B2: 0	B2: 3	**B2: G1**
						B3: 2	B3: 0	B3: 0	B3: 3	**B3: G1**
Dedifferentiated liposarcoma	3	0	<10	4	G2	B1: 3	B1: 0	B1: 0	B1: 4	**B1: G2**
						B2: x	B2: 100%	B2: x	B2: x	B2: x
Well-differentiated liposarcoma	1	<50%	<10	3	G1	B1: x	B1: 100%	B1: x	B1: x	B1: x
						B2: 1	B2: 0	B2: 0	B2: 2	**B2: G1**
						B3: 1	B3: 0	B3: 0	B3: 2	B3: G1
Myofibroblastic sarcoma	2	<50%	9	4	G2	B1: 2	B1: 0	B1: 0	B1: 3	B1: G1
						B2: 2	B2: 0	B2: 0	B2: 3	B2: G1
						B3: 2	B3: 0	B3: 0	B3: 3	B3: G1
Leiomyosarcoma	2	<50%	4	4	G2	B1: 2	B1: 0	B1:10	B1: 4	**B1: G2**
						B2: 2	B2: 0	B2: 2	B2: 3	B2: G1
						B3: 2	B3: 0	B3: 7	B3: 3	B3: G1
Myxofibrosarcoma	x	x	x	x	Gx	B1: x	B1: x	B1: x	B1: x	B1: Gx
						B2: x	B2: x	B2: x	B2: x	B2: Gx
Undifferentiated Sarcoma, NOS	3	<50%	1	5	G2	B1: 3	B1: 0	B1: 0	B1: 4	**B1: G2**
						B2: 3	B2: 0	B2: 0	B2: 4	**B2: G2**
						B3: 3	B3: <50%	B3: 0	B3: 5	**B3: G2**
MPNST	2	<50%	0	4	G2	B1: x	B1: 100%	B1: x	B1: x	B1: x
						B2: x	B2: 100%	B2: x	B2: x	B2: x
						B3: 2	B3: <50%	B3: 0	B3: 4	**B3: G2**
Undifferentiated Sarcoma, NOS	3	<50%	>20	7	G3	B1: 3	B1: 0	B1:16	B1: 5	B1: G2
						B2: 3	B2: 0	B2: 6	B2: 4	B2: G2
						B3: 3	B3: <50%	B3:15	B3: 6	**B3: G3**
Myofibroblastic sarcoma	2	0	12	4	G2	B1: 2	B1: 0	B1: 1	B1: 3	B1: G1
						B2: 2	B2: 0	B2: 0	B2: 3	B2: G1
						B3: 2	B3: 0	B3: 2	B3: 3	B3: G1
Solitary fibrous Tumor	1	0	2	2	G1	B1: 1	B1: 0	B1: 1	B1: 2	**B1: G1**
						B2: 1	B2: 0	B2: 1	B2: 2	**B2: G1**

TD: tumor differentiation; TN: tumor necrosis, MC: mitotic count; B: biopsy; x: score cannot be assessed due to extensive necrosis or regressive transformation of the tumor.

## Data Availability

The data presented in this study are available upon reasonable request from the corresponding author. The data are not publicly available due to ethical restrictions and data protection regulations.

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
