# Peer review of "Comparing Apparent Diffusion Coefficient and FNCLCC Grading to Improve Pretreatment Grading of Soft Tissue Sarcoma—A Translational Feasibility Study on Fusion Imaging"

_cancers, 2022, doi:10.3390/cancers14174331_

Round 1

Reviewer 1 Report

Dear Authors, 

Thank you for your work and interesting manuscript. As previous species have shown, the correct diagnosis of sarcoma on imaging is not easy but very important. The grading is one of the crucial points for the therapy. Even if imaging prediction of grading will never be superior to HIStological diagnosis in my opinion, it is still a helpful indicator. 

The number of included cases (n=12) with different tumor identities is in my opinion unfortunately a limitation of the study. Even though the design of a prospective study was chosen. 

Due to the small number of cases and the different tumor entities I would recommend to change the title of the study so that it is clear to the reader that it is a feasibility study (e.g.: .... - a translational feasibility study on fusion imaging). In addition, the title should be shortened. 

Figure 2 appears somewhat ambiguous. A clearer highlighting of the biopsy marker should be considered here. 

As a feasibility study, the study seems sufficient to me.

Author Response

Response to Reviewer 1 Comments

Thank you for your work and interesting manuscript. As previous species have shown, the correct diagnosis of sarcoma on imaging is not easy but very important. The grading is one of the crucial points for the therapy. Even if imaging prediction of grading will never be superior to Histological diagnosis in my opinion, it is still a helpful indicator.  The number of included cases (n=12) with different tumor identities is in my opinion unfortunately a limitation of the study. Even though the design of a prospective study was chosen. 

Point 1: Due to the small number of cases and the different tumor entities I would recommend to change the title of the study so that it is clear to the reader that it is a feasibility study (e.g.: .... – a translational feasibility study on fusion imaging). In addition, the title should be shortened. 

Response 1: The categorization as a feasibility study is correct. We have changed the title accordingly to “Comparing Apparent Diffusion Coefficient and FNCLCC Grading to Improve Pretreatment Grading of Soft Tissue Sarcoma – A Translational Feasibility Study on Fusion Imaging”.

Point 2: Figure 2 appears somewhat ambiguous. A clearer highlighting of the biopsy marker should be considered here. 

Response 2: To improve the visibility of the biopsy markers in figure 2 we highlighted their position by arrows. 

As a feasibility study, the study seems sufficient to me.

Response 3: Thank you for your comments and advice. In addition to your written comments, you suggested moderate English changes. For language improvement, editing by an expert from www.scribbr.com was performed now.

Reviewer 2 Report

I would say that the topic addressed in this study is of clinical significance because it addresses a problem in the clinical practice of soft-tissue sarcomas. On the other hand, I regret to point out that there are major problems with the study design, which I would find very disappointing.

The hypothesis that the authors wanted to clarify in this study would be whether the pre-treatment grading of soft-tissue sarcomas can be made more accurate by taking MRI findings into account. As the authors mention, FNCLCC grading in biopsy specimens is highly problematic in terms of its accuracy.

In the present study cohort, it is likely that biopsies are indeed performed preoperatively for real. In a "real" needle biopsy, not in ex vivo biopsy, multiple samples are commonly taken. I would think that the simplest and best method is to look at the grade of the "real" biopsy specimen, the grade that takes into account the MRI findings proposed by the authors, and the concordance rate between these and the grade of the resected specimen.

In addition, in this study, only 12 patients were analyzed, an overwhelmingly insufficient number. Strictly speaking, well-differentiated liposarcoma and solitary fibrous tumor are not "malignant" in the WHO classification and are not included in the FNCLCC classification. Given this, the number of valid analyses is only 10, and it would be impossible to draw any conclusions from an analysis of only 10 cases including a variety of histologic types.

Also, was the analysis of dedifferentiated liposarcoma performed on both the dedifferentiated and well-differentiated components? Or was the analysis of the dedifferentiated component only?

For these reasons, I would recommend a fundamental revision of the study plan, including a major modification of the study design.

Author Response

Response to Reviewer 2 Comments

I would say that the topic addressed in this study is of clinical significance because it addresses a problem in the clinical practice of soft-tissue sarcomas. On the other hand, I regret to point out that there are major problems with the study design, which I would find very disappointing. The hypothesis that the authors wanted to clarify in this study would be whether the pre-treatment grading of soft-tissue sarcomas can be made more accurate by taking MRI findings into account. As the authors mention, FNCLCC grading in biopsy specimens is highly problematic in terms of its accuracy.

Point 1: In the present study cohort, it is likely that biopsies are indeed performed preoperatively for real. In a "real" needle biopsy, not in ex vivo biopsy, multiple samples are commonly taken. I would think that the simplest and best method is to look at the grade of the "real" biopsy specimen, the grade that takes into account the MRI findings proposed by the authors, and the concordance rate between these and the grade of the resected specimen.

Response 1: Thank you for your suggestions. You proposed to use routine clinical data to compare histological and radiographic grading (e.g. as a prospective or even retrospective study). As discussed in the manuscript, this approach was tried before and not successful. The accuracy of identifying high-grade tumors by imaging features taken from routine procedures was reported to be only between 64% and 83.0%.

There are several reasons why study designs evaluating routine biopsies are limited:

  • Functional imaging is not a routine procedure. If patients are referred from non-specialized centers and general quality of anatomic imaging is acceptable, MRI is not repeated in clinical routine at most sarcoma centers. These patients are not available for further analysis, limit the number of patients and probably lead to selection bias.
  • Biopsy localizations cannot be determined exactly in clinical routine. Routine biopsies of soft tissue masses are performed under ultrasound or CT guidance in a fan-shaped manner. Localization of each single biopsy is neither performed nor feasible – especially if ultrasound guided biopsies are performed. A site-specific correlation of radiographic and pathologic features is not feasible.
  • Whole tumor ADC from routine functional MRI (if available) may be chosen to determine radiographic grading as an alternative. Since soft tissue sarcomas are frequently large (>5cm) and strongly heterogeneous it is questionable if whole tumor ADC (which would include necrotic areas etc.) is the best candidate for radiographic grading. Determination of minimal ADC may be more representative and may correlate better with histopathological grading.

We considered two other options for the design of this study:

  • Placing radiopaque markers at the biopsy site - analogous to those used in biopsies of breast cancer. However, these markers are not licensed for soft tissue sarcoma in Germany. Furthermore, additional post-interventional imaging for image fusion would also have been required.
  • Performing CT-guided biopsies in all patients. Subsequently, the biopsy tracts (or the position of the coaxial needle) could have been correlated with preexisting functional imaging.

Please note, that both alternatives would have required feasibility studies before conducting a large-scale trial since preliminary data do not exist and calculations for costs and number of patients as well as selection of study centers would not have been reasonable. For both alternatives, the development of fusion imaging would have been necessary to correlate biopsy site and functional imaging. For the second alternative design involving CT-guided biopsies, radiation exposure would have been increased - at least in comparison to our own routine approach - in which ultrasound-guided biopsies are preferred. An additional advantage of the proposed study design was that we were able to target almost all tumor regions. This is usually not possible in a large proportion of soft tissue lesions during routine biopsy procedure due to the proximity of critical structures (e.g. nerves, intestine).

Point 2: In addition, in this study, only 12 patients were analyzed, an overwhelmingly insufficient number. Strictly speaking, well-differentiated liposarcoma and solitary fibrous tumor are not "malignant" in the WHO classification and are not included in the FNCLCC classification. Given this, the number of valid analyses is only 10, and it would be impossible to draw any conclusions from an analysis of only 10 cases including a variety of histologic types.

Response 2: The current WHO classification classifies both tumors as malignant in dependence of further characteristics (e.g. localization and size). (WHO classification of Tumours Editorial Board. Soft Tissue and Bone Tumours, edited by the WHO Classification of Tumours Editorial Board. Fifth edition. 2020. ISBN 9789283245025. Page 2, 1st column, line 26 and 27 and 2nd column, line 21 and 22.)

Point 3: Also, was the analysis of dedifferentiated liposarcoma performed on both the dedifferentiated and well-differentiated components? Or was the analysis of the dedifferentiated component only?

Response 3: Both, dedifferentiated and well-dedifferentiated components were analyzed.

Point 4: For these reasons, I would recommend a fundamental revision of the study plan, including a major modification of the study design.

Response 4: Thank you for your comments and suggestions. We added a critical discussion of the study design in the revised manuscript (page 11, line 338 to 364 of the revised manuscript). As outlined above, we had valid reasons to select the proposed study design. We are aware of its limitations and the necessity of further trials to prove the concept of radiographic imaging. We appreciate your critical comments that will be useful to develop a subsequent prospective large-scale trial. Regardless of its design, our feasibility study may contribute to perform such a large-scale trial in a rare disease (e.g. in calculating the number of patients, convincing participating sarcoma centers and funding sources, harmonizing functional imaging, pathological evaluation and improving fusion imaging). We therefore consider the publication of the current results as appropriate.

Round 2

Reviewer 2 Report

The authors wrote, "The current WHO classification classifies both tumors as malignant in dependence of further characteristics (e.g. localization and size). (WHO classification of Tumours Editorial Board. Soft Tissue and Bone Tumours, edited by the WHO Classification of Tumours Editorial Board. Edition. 2020. ISBN 9789283245025. Page 2, 1st column, line 26 and 27 and 2nd column, line 21 and 22."

If the authors are correct that the case analyzed here was of deep trunk origin, such as retroperitoneal, and that was definitely well-differentiated liposarcoma rather than atypical lipomatous tumor, then I think it can be included in the analysis. This point needs to be mentioned.

Also, regarding solitary fibrous tumor (SFT), were the pathological findings diagnostic of malignant SFT? As the authors say, a malignant SFT is classified as malignant, but a normal SFT would not be classified as malignant. I would appreciate further additional clarification on this point.

Author Response

The authors wrote, "The current WHO classification classifies both tumors as malignant in dependence of further characteristics (e.g. localization and size). (WHO classification of Tumours Editorial Board. Soft Tissue and Bone Tumours, edited by the WHO Classification of Tumours Editorial Board. Edition. 2020. ISBN 9789283245025. Page 2, 1st column, line 26 and 27 and 2nd column, line 21 and 22."

If the authors are correct that the case analyzed here was of deep trunk origin, such as retroperitoneal, and that was definitely well-differentiated liposarcoma rather than atypical lipomatous tumor, then I think it can be included in the analysis. This point needs to be mentioned.

The tumor was located deep to the fascia, had a size of 20x14x10 cm, originated within the adductor muscles and grew into the ischiocrural muscles. In addition to these clinical criteria, imaging and pathology revealed necrotic areas within the tumor. Resection with sparing of the sciatic nerve (R1) and radiation therapy were administered postoperatively (compare page 5, lines 165 - 169).

Also, regarding solitary fibrous tumor (SFT), were the pathological findings diagnostic of malignant SFT? As the authors say, a malignant SFT is classified as malignant, but a normal SFT would not be classified as malignant. I would appreciate further additional clarification on this point. 

The SFT was located deep to the fascia, had a size of 6.2 cm, and originated in the fossa popliteal. It was classified to have a low risk of recurrence (compare page 5, lines 169 - 171). 

Several scoring systems have been established for SFT. They consider clinical and histopathological factors, including the proportion of necrotic tumor tissue and the mitotic count. Recently, three SFT scoring systems were evaluated in a large validation study involving more than 300 SFT patients (Georgiesh et al. 2022, British Journal of Cancer, doi:10.1038/s41416-022-01959-4.). All scoring systems were able to differentiate between high- and low-risk cases. Regardless of the scoring system, all low-risk cohorts included patients with recurrences. Consequently, there was no zero-risk group in any of the scoring systems.

The aim of this study was to evaluate the feasibility of fusion imaging and to assess the potential of diffusion weighted imaging to differentiate between low- and high-risk lesions. An exclusion of SFT patients from the analysis is not justified here. Irrespective of different cut-offs in mitotic count and necrosis, the tumor was low-risk/low-grade according to any SFT and the FNCLCC scoring system. The inclusion of these types of low-risk SFTs in future trials will be essential to learn how to differentiate them from high-risk tumors.

We added a comment in the discussion concerning the importance of including subtype-specific scoring systems in the risk assessment of STS in subsequent studies (page 10, lines 319 - 322).